

# Estimating the thickness of unconsolidated coastal aquifers along the global coastline

Daniel Zamrsky[1], Gualbert H.P. Oude Essink[1,2], and Marc F.P. Bierkens[1,2]

[1]Department of Physical geography, Utrecht University, Utrecht, The Netherlands
[2]Deltares, Utrecht, The Netherlands

*Correspondence to*: Daniel Zamrsky (d.zamrsky@uu.nl)

**Abstract.** Knowledge of the thickness of aquifers is crucial for setting up numerical groundwater flow models in support of the management and control of groundwater resources. Fresh groundwater reserves in coastal aquifers are particularly under threat of salinization and depletion as a result of climate change, sea-level rise, and excessive groundwater withdrawal under

urbanization. To correctly assess the possible impacts of these pressures we must have better information about subsurface conditions in coastal zones. Here, we propose a method that combines available global datasets to estimate, along the global coastline, the thickness of aquifers formed by unconsolidated sediments. To validate our final estimation results, we collected both borehole and literature data. Additionally, we performed a numerical modelling study of the effects of varying aquifer thickness and geological complexity on simulated saltwater intrusion. The results show that our aquifer thickness

estimates can indeed be used for regional scale groundwater flow modelling but that for local assessments additional geological information should be included. The final dataset can be downloaded via https://doi.pangaea.de/10.1594/PANGAEA.880771.

## 1 Introduction

Coastal aquifers provide the source of fresh groundwater for more than two billion people worldwide, (Ferguson & Gleeson,

2012). Multiple local and regional studies have shown that these fresh groundwater resources are not only threatened by natural disasters such as storm surges and tsunamis (Cardenas et al. 2015), but also increasingly by climate-induced sea level rise (Carretero et al. 2013; Rasmussen et al. 2013; Sefelnasr et al. 2014) and urbanization that leads to over-exploitation of coastal aquifers combined with reduced groundwater recharge (Custodio 2002).

Comparing vulnerabilities of coastal aquifers worldwide in a consistent manner requires a global scale study (Döll 2009).

However, many of the necessary input datasets, both physical and societal, are only available on regional or local scale and can therefore only be used in regional investigations of coastal aquifer vulnerability. Notable work on global vulnerability of coastal aquifers are studies by Ranjan et al. (2009), and Michael et al. (2013) looking at coastal aquifer vulnerability to sea water intrusion and by Nicholls & Cazenave (2010) taking social-economic factors into account. Related studies are assessments of the global occurrence of saline groundwater (Van Weert, et al. 2008) and the existence of offshore fresh or





brackish groundwater (Post et al. 2013) based on observational data. Due to lack of global information, the few studies that attempted a modelling approach (i.e. Ranjan et al. (2009) and Michael et al. (2013)) used globally or regionally homogenous hydraulic parameters. Indeed, recent reviews concluded that most of the past modelling studies until present day (both on local and global scale) considered a homogeneous aquifer system (Werner et al. 2013; Ketabchi et al. 2016). This pinpoints

that there is still a large gap in our knowledge about the hydrogeological setting of coastal aquifers worldwide. Since the local and regional hydrogeological conditions largely determine the vulnerability of coastal aquifers to sea level rise (Michael et al. 2013) and groundwater pumping (Ferguson & Gleeson, 2012), it is important to improve our insight into the hydrogeological characteristics of coastal aquifers.

The goal of this study is to estimate the thickness of coastal aquifer systems along the global coastline. This constitutes a first

step towards a more complete hydrogeological characterization of coastal aquifers. Our focus is limited to aquifer systems formed by unconsolidated sediments and uses only available open source global datasets during the estimation process. The dataset thus created is different from previously created global datasets (Table S1 in the Supplementary Information) that either focused only on estimating the thickness of the soil (or regolith) layer, (Pelletier et al. 2016; Shangguan et al. 2017), or estimate the total thickness of porous media and do not distinguish between unconsolidated and consolidated sediments or

rocks (Whittaker et al. 2013; de Graaf et al. 2015). To illustrate the use of the new dataset in a regional groundwater modelling, we will show the results of variable-density groundwater flow and coupled salt transport models for three distinctly different coastal cross-sections. We also show the sensitivity of modelling results to varying the aquifer thickness and geological complexity.

## 2 Materials and Methods

### 20  2.1 Sediment thickness estimation

We collected state of the art open source global datasets (Table 1) that provide information on topography and bathymetry (Weatherall et al. 2015), thickness estimation of the surface sediments (Pelletier et al. 2016), aquifer thickness estimation from a global hydrological model (de Graaf et al. 2015), lithology (Hartmann & Moosdorf 2012) and coastline position (Natural Earth 2017). The core of our aquifer thickness estimation (ATE) method is to combine topographical and

lithological information. This enables us to find the topographical slope of outcropping bedrock formations and to determine the extent of the coastal plain. The latter is defined by a low topographical slope (Weatherall et al. 2015), a lithology consisting of unconsolidated sediments (Hartmann & Moosdorf 2012) and a thickness of subsurface sediments (regolith) of more than 50m (Pelletier et al. 2016). This is the first study that directly combines lithology and topographic information to estimate the thickness of unconsolidated sediment formations on global scale.

Given the large variety of coastal environments ranging from steep cliffs to extensive deltaic flat areas, it is important to develop a robust method that distinguishes between these different coastal types and also take into account variations of inland bedrock formations. To achieve this, the coastal zones are represented as perpendicular cross-sections to the coastline



and are placed equidistantly (5km) along the coastline. The intersections between the cross-sections and the coastline are called coastal points. Along the cross-section, a set of equidistant points (0.5km) are positioned (cross-section points) and mark the locations where values from the datasets listed above are extracted (Fig. 1a). The cross-sections span 200 km both inland and offshore from the coastal point to capture the bathymetrical and topographical profile. This distance was chosen

to safely cover the necessary stretch both landward and offshore for groundwater flow and coupled salt transport modelling. Recent studies dealing with the latter set the landward boundary less than 200km from the coastline even in deltaic areas (Delsman et al. 2013; Larsen et al. 2017; Nofal et al. 2016). Similarly, previous studies showed that submarine groundwater discharge can occur more than 100km offshore (Kooi & Groen 2001; Post et al. 2013).

Figure 1b shows an example of a cross-section running through a coastal point. All the necessary values from the individual
datasets are aggregated and used to determine the extent of the coastal plain and the position of the anchor point. The inland boundary point of the coastal plain is defined as a cross-section point that is of a lithological class different than a water body (to take into account e.g. lagoons, bays) or unconsolidated sediments based on the GLIM dataset.

Once the extent of the coastal plain is known, the next step is to define the anchor point using the soil and sedimentary deposit thickness dataset (Pelletier et al. 2016). Taking note of the fact that the Pelletier et al. dataset generally has increasing
thickness values towards the coast in case of unconsolidated sediments and that a thickness larger than 50m is not mapped, we define the anchor point as the last cross-section point (moving from land to coast) with soil and sedimentary deposit thickness smaller than 50m. This anchor point represents the last point of known soil and sedimentary deposit thickness and is located below ground at the indicated depth by this dataset. The ATE is then performed for all cross-section points located between the anchor point and the coastline.

Due to a large variety in the extent of the coastal plain, topography and geological diversity of the coastal cross-sections worldwide, four different estimation methods are proposed to increase the robustness of the overall estimation method. The differences are in the selection of topographical points that are used to simulate the bedrock slope (Fig. 1c). The anchor point is added to the set of topographical points in every estimation method.

The first method selects all cross-section points elevation values of the first peak located prior to the coastal plain, regardless
of the lithological class. The second method selects all the cross-section points elevation values of the highest peak located in a bedrock formation (any different class than unconsolidated sediments). The third selection method consists of selecting all cross-section points elevations located between the end of the coastal plain and the end of the bedrock formation. The last method selects only the local minimum and maximum points of every peak located behind the coastal plain. This diversity in selecting cross-section points based on combinations of lithological and topographical information allows for a more robust
method that is fit for various coastal environments.

For each of the selection methods described above, a curve-fitting of first and second order is performed to simulate the slope of the bedrock formation (Fig. 1c). If the minimum point of the second order curve is situated before the coastline we use three different types of linear functions to extend the bedrock slope simulation and estimate the sediment thickness by extending it beyond the coastline. All the three lines start at the minimum point of the second order estimation and run





through the coastline. The first line is a constant horizontal line, the second simulates the average slope of the continental shelf (defined as shallower than -200m. bsl.) and the last line simulates the average slope of the whole 200km offshore segment of the cross-section.

The estimated thickness provided by the PCR-GLOBWB dataset is chosen as the lower boundary since it tends to

overestimate the coastal thickness because its underlying method is more fit to the inland areas and uses river networks and basins as basis for thickness estimates (de Graaf et al. 2015). Finally, all the points are used to estimate the mean, minimum and maximum thickness of the aquifer at the coastline and the mean coastal profile for the extent of the unconsolidated sediments. The dataset that is stored contains per coastal cross-section the mean profile as well as the maximum and minimum depth and the standard deviation of the depth at the coastline. For each coastal cross-section, also the position and

depth of the anchor point is included.

## 2.2 Validation methods

Two different validation approaches are applied to test the fit of our estimated aquifer thicknesses with measured values. First, the results are compared with information from available open source geological borehole datasets. The second validation method consists of comparing the average estimated aquifer thickness with measured values gathered via a

literature review.

A dataset consisting of 112 geological borehole descriptions was collected and sorted out from open source datasets and web services, mostly located in Brazil and Australia. After digitizing the borehole reports, we translated the geological information to overall unconsolidated sediment thickness to compare it to our final thickness estimates. This means that all the unconsolidated sediment types such as sand, clay or silt were merged into the same stratigraphic unit and their overall

thickness is taken as the final sediment thickness. Figure 2 shows the location of the collected borehole data, the sources of the data are presented in the supplementary information (Table S1). Since some of the boreholes are not located in direct proximity to the coastline, we chose to extrapolate the estimated sediment thickness by calculating the estimated sediment thickness for each cross-section point. This was done by creating a line between the anchor point depth and the estimated depth of the sediments at the coastline (Fig. 3). Next, the average thickness of the cross-section points in a circle with radius

of 2.5km around the borehole is compared to the thickness in the borehole.

The final literature validation set is composed of maximum, minimum and/or average aquifer thickness values (unconsolidated sediment) of 64 coastal areas worldwide. However, not all the literature sources provide the average unconsolidated sediment thickness. In the cases where it does not, it is calculated as half the maximum indicated thickness in case only the maximum value is provided. If both maximum and minimum thicknesses are given, the average thickness is set

to be halfway between these two values. The table with references of the literature sources and the sediment thickness values provided by these sources are listed in the supplementary information (Tables S2 and S3). The final estimated average sediment thickness values were compared with the literature dataset and evaluated based on the relative error percentage and



relative improvement compared to the overall average thickness value from all literature sources. The relative error percentage is based on the following equation:

$$RE = 100 * \frac{Z_{est} - Z_{lit}}{Z_{lit}} \qquad (1)$$

where $Z_{est}$ is the ATE by our method and $Z_{lit}$ is the average thickness given by literature.

The overall average thickness value based on all literature sources was calculated using the equation below:

$$Z_{lit_{avg}} = \frac{1}{N} \sum_{i=1}^{n} Z_i \qquad (2)$$

where $Z_{lit_{avg}}$ is the overall average value of all literature values $Z_i$.

The mean absolute error was then calculated for both the overall average value and the estimated average thickness values suggested by our method, see equations below:

$$MAE_{lit} = \frac{1}{N} \sum_{i=1}^{n} |Z_{lit_{avg}} - Z_i| \qquad (3)$$

$$MAE_{est} = \frac{1}{N} \sum_{i=1}^{n} |Z_{est} - Z_i| \qquad (4)$$

Subsequently, the relative improvement rate is calculated as follows:

20

$$RI = 100 * \frac{MAE_{lit} - MAE_{est}}{MAE_{lit}} \qquad (5)$$

The same validation criteria are calculated using the borehole data.

**2.3 Groundwater flow and salt transport modelling**

25 A set of variable-density groundwater flow models with varying aquifer thickness is created to investigate the effects of aquifer thickness and geological complexity on the concentration profile and the total volume of fresh groundwater in the coastal zone. These models were set up for the three example cross-sections mentioned in section 2.1 using the SEAWAT code (Guo & Langevin 2002) and the Python Flopy library (Bakker et al. 2016). The model schematizations and list of input parameter values is listed in the supplementary information (Fig. S3 and Table S4).

30



## 3 Results

### 3.1 Sediment thickness estimation

The aquifer thickness is estimated for a total of 26 968 coastal points around the globe, which covers roughly one fifth of the global coastline. The rest of the global coastline is covered by other lithological types than unconsolidated sediments and is

not taken into account by the ATE method. The overall ATE results are presented in Figure 4a. It shows that the aquifer thickness estimates range between 0.1m and more than 5000m, with mean value close to 170m. In total 87% out of all the ATEs predict a thickness lower than or equal to 300m (see Figure 4b). A similar result is observed in the analysis of the literature sources, where more than 65% of the studied areas have aquifer thickness lower than or equal to 300m. This difference is explained by the fact that a disproportionally large number of deltaic areas with thick sediment layers is

included in the literature review.

### 3.2 Validation of ATE results

When comparing our ATE with the information collected from the borehole dataset, it is clear that our ATE method provides estimates right order of magnitude, but it cannot capture local variations of aquifer thickness. Figure 5a shows that the majority of ATE have (absolute) relative error values (Equation 1) lower than 100%, meaning that our results are in the same

order of magnitude as observed values from the borehole dataset. However, the relative improvement of the ATE, as compared with using the average of the borehole thicknesses as estimate (Equation 5) is inconclusive as the amount of positive values is nearly equal to the total of negative values (Figure 5b).

The results of the validation with the coastal sediment thickness values gathered via a literature review show a more positive result compared to the borehole validation. The overall average thickness of the literature dataset is 353m, while about 69%

of all studied areas have a sediment thickness of 300m or lower. The relative improvement of sediment thickness estimates using our method is about 22% compared to the overall average of the thickness values as indicated by the literature (see supplementary information). However, in coastal zones that have the average sediment thickness of 300m or less, the relative improvement of our method is around 59%. Since our results suggest that 87% of the global coastline that is composed of unconsolidated sediments has average thickness of 300m or lower, the higher relative improvement achieved by our method

gains extra importance.

Overall, about 48% of the validation areas have the absolute relative error percentage below 50%, while 35% of validation areas have the absolute relative error percentage between 50% and 100% (Fig. 5c). Still, 17% of the validation areas show absolute relative error percentage higher than 100%. A closer look at Figure 3c reveals that the majority of these validation areas have the average thickness (based on literature) lower than 100m. However, the overall results for validation areas with

average thickness lower than 300m show that 59% have relative error percentage lower than 50%, this is a 11% increase compared to the overall validation dataset.



### 3.3 Groundwater flow and salt transport modelling

The main goal of the numerical modelling performed in our study is to evaluate the effect of variations in aquifer thickness values at the coastline on the salinity profile in coastal groundwater bodies. This can be done by comparing the salinity profiles of all simulations at a fixed time or by comparing the percentage of fresh water cells in the coastal zone. Apart from

variation of the thickness values, we also implemented a more complex geological scenario for each of the three test cases along with a homogenous geological one. This was done to evaluate the relative importance of aquifer thickness to the effect of geological complexity.

Figure 6 presents a sample of simulated salinity profiles for selected aquifer thickness values for the three test cases. The complete set of the simulated salinity profiles together with the model conceptualization and model parameters and variables

is given in the supplementary information (Fig. S3 to S6 and Table S5). While comparing the salinity profiles for different aquifer thicknesses it is apparent that variations of aquifer thickness for homogenous geological conditions (Figs. on the right) do not have large effects on the fresh-saline distribution, except for the lowest aquifer thickness value, (Figs. 6a, 6c). Figure 6b shows that the thicker the aquifer at the coastline, the more saline water intrudes inland and in some cases upconing under low lying areas can be observed (Fig. 6c).

The implementation of complex geological conditions based on the literature description that existed about these sites consisted mainly of introducing low conducting layers (aquitards). As Figure 6 shows, an aquitard has a substantial effect on the final salinity profile when compared to the salinity profile for homogenous geological conditions with the same aquifer thickness. The position of the aquitard combined with varying aquifer thickness has a large effect on the salinity profile and potential fresh (or brackish) groundwater offshore reserves (Fig. 6b left column). In particular the simulations with larger

aquifer thickness values show fresh (or brackish) offshore groundwater below the aquitard layer. Similar patterns can be observed in the last test case (Fig. 6c), where the aquitard layers prevent saline water from intruding inland and show large volumes of offshore brackish water.

Comparison of fresh groundwater cells percentage within the coastal zone of all three test cases (Fig. 6) shows a trend where geological scenario (homogenous versus complex) has a larger effect on the amount of estimated fresh groundwater reserves

than varying aquifer thickness. The largest differences of fresh groundwater cells percentages for the same geological scenario can be observed, in most cases, between the extreme values of aquifer thickness (lowest vs. highest).

### 4. Discussion and conclusion

Although in the right order of magnitude (Figure 5a), the validation of the ATE with borehole measurements is worse than

those compared with reported values in the literature. The large scale-discrepancy between our global ATE dataset and boreholes is the most obvious cause for this. It shows that our approach is not detailed enough to estimate very local variations in aquifer thickness as picked up by boreholes. Boreholes will generally lie between profile locations, which




means that local variation also results in spatial dislocation errors, even though spatial averaging is used to bridge the scale gap (Fig. 3). Still, even when compare to boreholes, we observe an overall improvement of the ATE method performance for coastal areas with measured thickness between 100m and 300m. The comparison between literature values comes out more favourably, because the synthesis of data in the form of spatial statistics and geological profiles is a form of spatial

aggregation that better matches the scale of our estimation method. We have used the validation data that could be collected during the course of this study, but the validation set is far from exhaustive. The validation dataset should be expanded and continuously improved to achieve better ATE along the global coastline.

Our method tends to underestimate the aquifer thickness in deeper systems, such as large complex deltaic sedimentary structures with measured average aquifer thickness larger than 500m (Fig. 5). This could be due to the limited cross-section

length that spans at most 200km inland and offshore from the coastline depending on the extent of the coastal plain. If the latter exceeds this maximum length then no bedrock formation is found and thus no aquifer thickness is estimated. In case the bedrock formation is only partially taken into account (e.g. only the foothill of a mountain range), its topographical slope will be lower which leads to lower ATE values at the coastline. The opposite happens for coastal areas with measured average thickness lower than 100m. In these cases, our average ATE values tend to be overestimated (Fig. 5). This could

again be caused by the resolution of the input dataset which creates larger errors on local scale and for shallow systems which by themselves have a smaller size than more extensive coastal plains.

The numerical modelling results show that only the simulations with extreme ATE values give substantially different results from the simulations with average or close to average ATE values. More variation in the fraction of fresh groundwater cells in the coastal zone can be observed in the test case with intermediate aquifer thickness (Fig. 6b). In the other two test cases

(Figs. 6a and 6c) the variation in the fraction of fresh groundwater cells is very low for both geological scenarios. On the other hand, the model results also show that geological complexity (multi-layering) has a big impact on the results. Thus, for locally meaningful results, the aquifer thickness is but a first result, and a global estimate of multi-layering (aquifers and aquitards) is a necessary next step.

When comparing our numerical modelling output (with the complex geology incorporated) with the salinity profiles reported

from the individual studies (Pranzini 2002, Yechieli et al. 2010 and Trapp & Horn 1997) we find that differences for the cases a) and b) are small and a 2D schematization suffices. However, for cross-section c) the differences are considerable. This is most likely due to the presence of strong alongshore flows in the area, a more complex upper hydrological system and the distribution of groundwater withdrawals in the area. This shows that 2D-approach modelling approach does not always suffice to estimate coastal groundwater flow.

In conclusion, we showed that it is possible to obtain first order estimates of coastal aquifer thickness by using available global datasets and a simple methodology consisting of simulating the bedrock slope from the geological outcrops. These estimates should be used for regional or global scale studies and are not suitable for detailed local models, in which case additional local geological information should be included. An important conclusion is that at the local scale geological complexity seems to play a larger role in simulated salinity concentration profiles than aquifer thickness (except for extreme



values). Thus, our ATE dataset provides a satisfactory first step towards a global coastal aquifer characterization that should be followed by the assessment of the geological complexity of coastal aquifers for local application.

## 5. Data format and availability

The final output data provides both the ATE at the coastline and the location and depth of the corresponding anchor points.
These data are given as shapefile and comma separated value files. The data can be downloaded via https://doi.pangaea.de/10.1594/PANGAEA.880771.

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



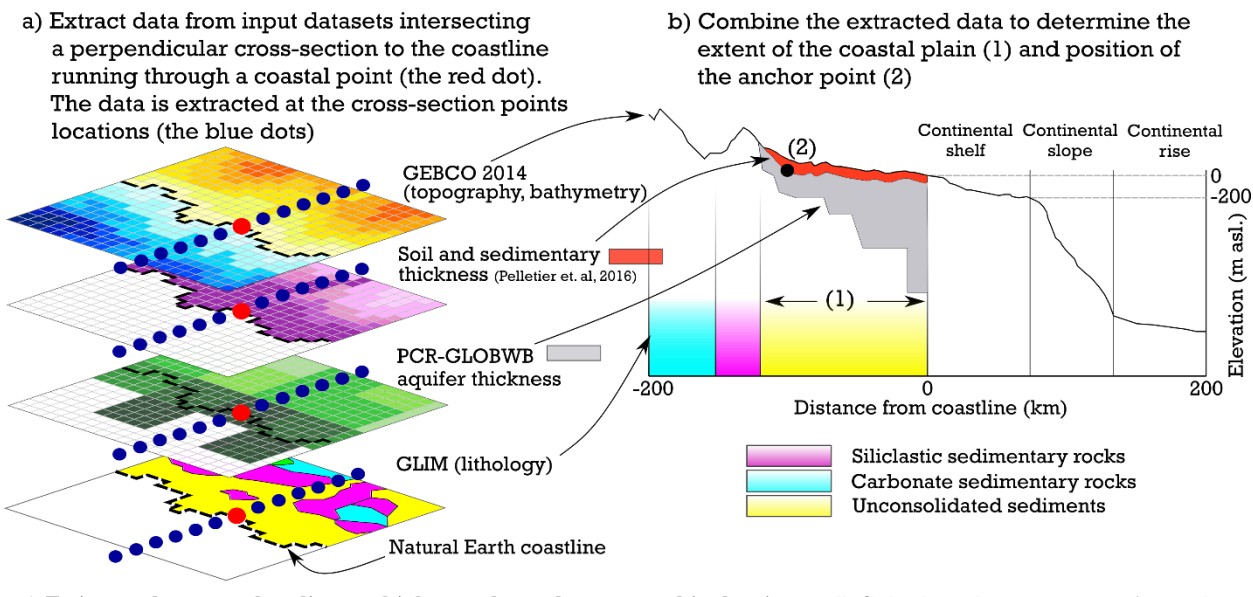

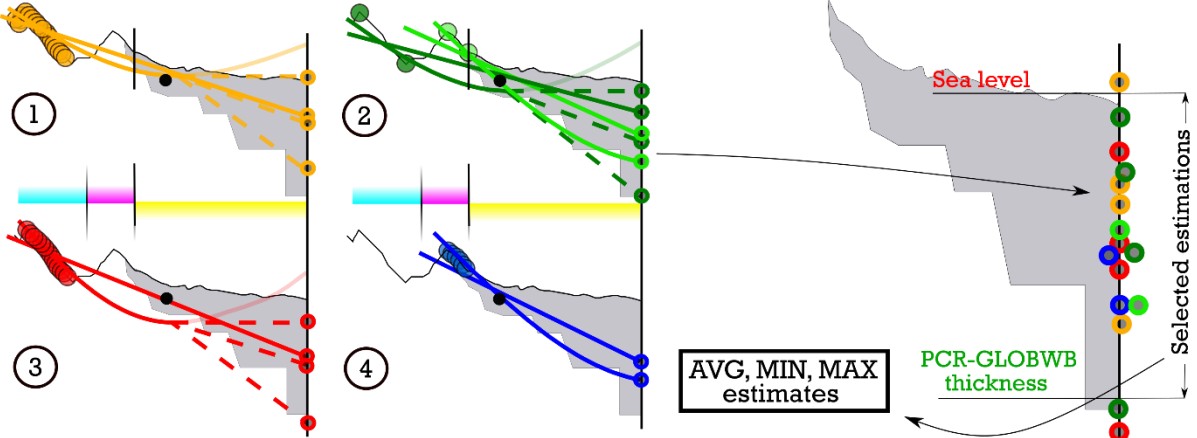

**Figure 1 Schematisation of the ATE method using available open source global datasets. (a) Combining datasets and extracting the values at cross-section points, only few are schematized in the figure (in reality 800 per cross-section). (b) Extent of the cross-section is set to 200km landward and offshore, (c) Various ATE lines, the 2nd order estimation line is not used for estimation in case its minimum is reached before the coastline (transparent). (d) Final step of calculating the average, minimum and maximum estimated values.**




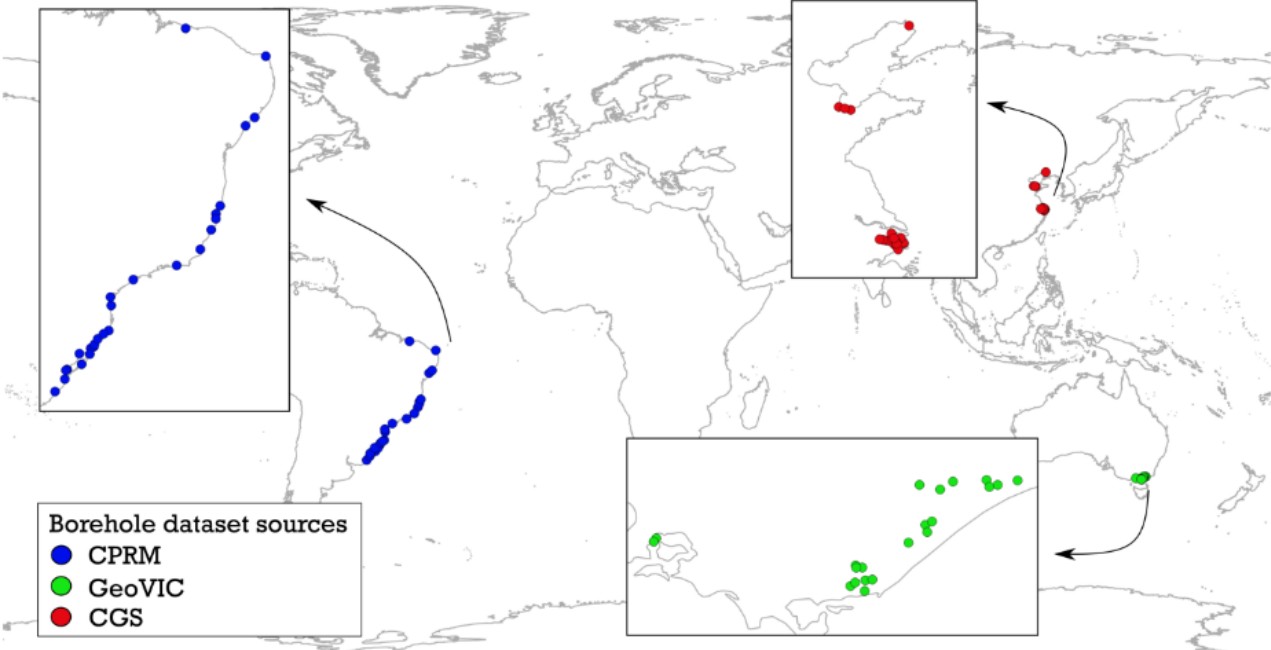

**Figure 2 Location of the borehole data used as validation dataset, sources are listed in the Table S3. The borehole information in Brazil and Australia was manually digitized while the subsurface information in China was gathered by interpreting the cross-section provided in the hydrogeological maps.**





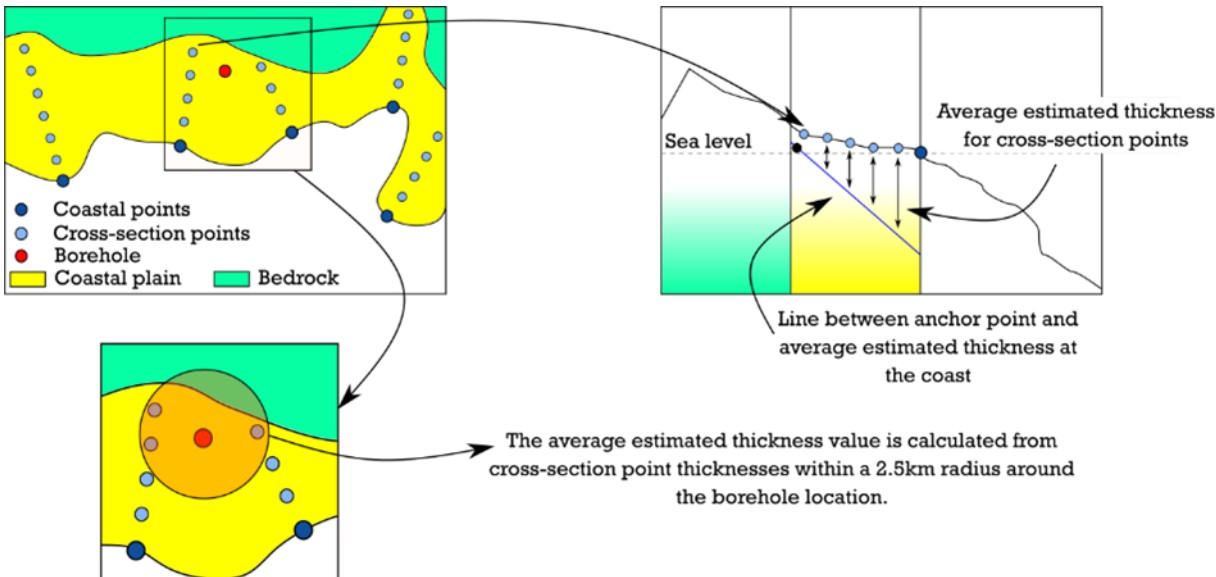

**Figure 3 Schematization of the borehole validation process. A set of points lying within a given distance is selected for each borehole and their estimated sediment thickness is averaged. The final comparison between these average values and measured values from the boreholes is shown in Figure 3 in the main article.**

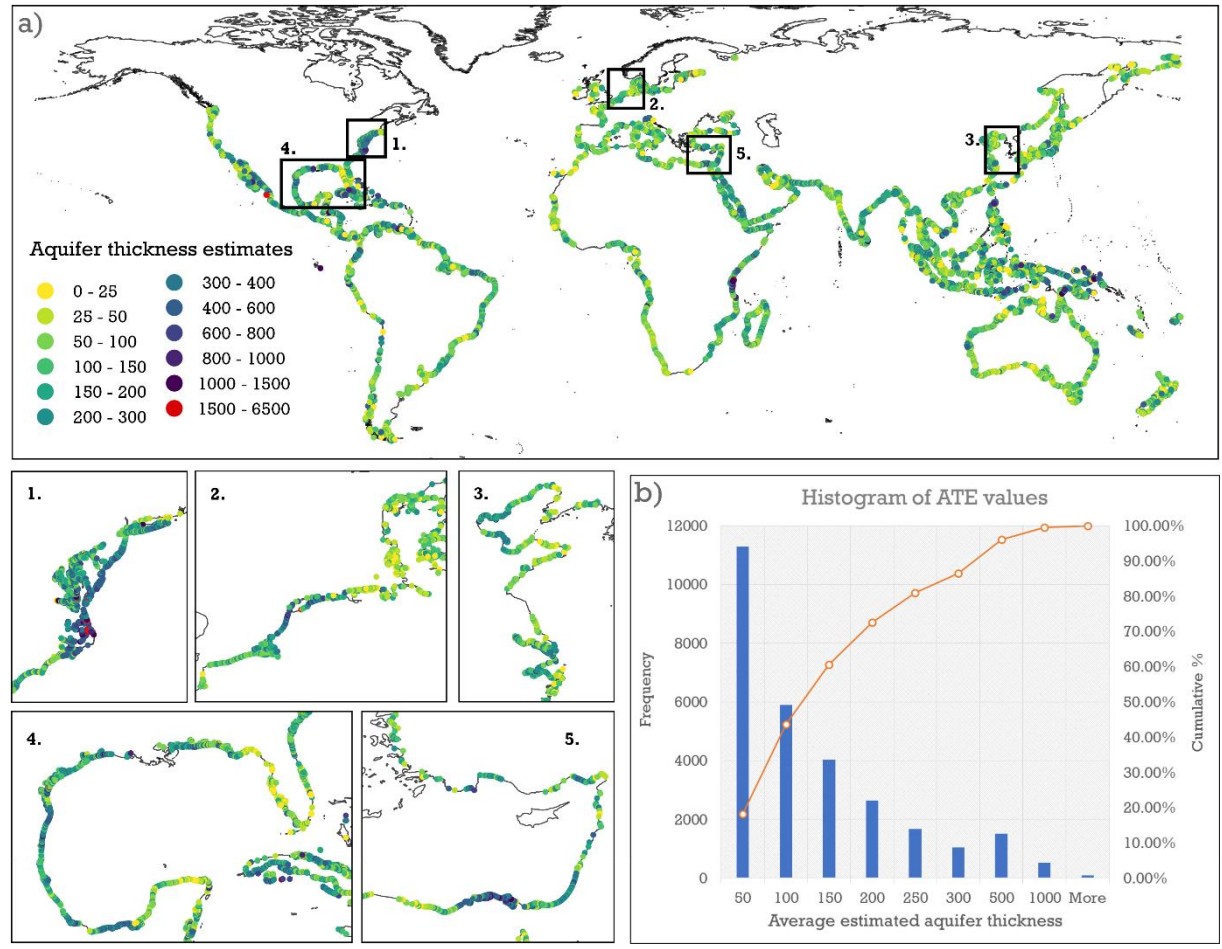

**Figure 4 (a) Global map of ATE at the coastline and zoomed areas (1-5) showing regional variations of estimated thickness in various coastal zones around the world. The coastal points are magnified giving the impression that more than the stated 20% of the global coastline is covered, which is not the case (see plain black line). (b) histogram of ATE values with cumulative frequency in %.**





**Figure 5 Overall borehole (a), (b) and literature (c), (d) validation results of the ATE results.**



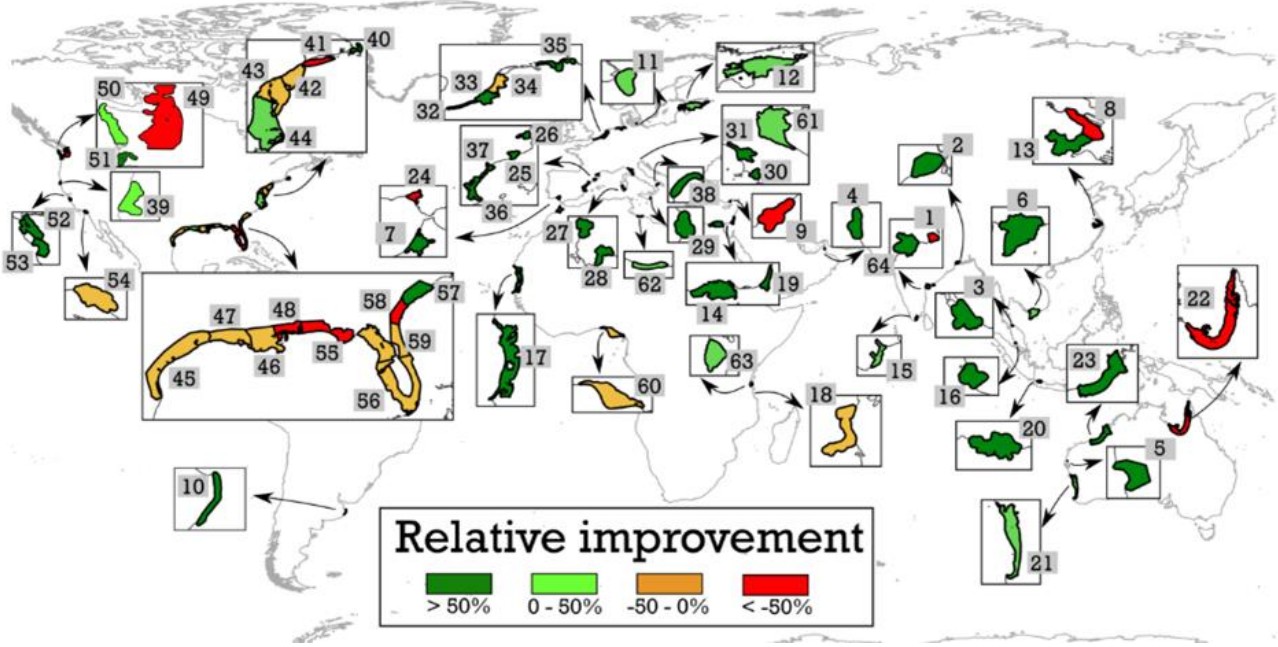

**Figure 6** Relative improvement of our estimated sediment thickness compared to using the overall average thickness value from all literature sources.





**Figure 7** Simulation results as salinity concentration profiles for the cross-sections located in the a) Versilia plain, Italy, b) Mediterranean aquifer, Israel and c) Virginia, USA with varying aquifer thickness and 2 geology scenarios. The local geological information for each area (a) (Pranzini 2002) (b) (Yechieli et al. 2010) (c) (Trapp Jr. & Horn 1997) was implemented (left column) together with homogeneous aquifer system (right column) to investigate the effects of geological complexity and aquifer thickness on simulated salinity profiles.



**Table 1 Summary of global datasets used for aquifer thickness estimation.**

| Dataset name | Description | Resolution | Reference |
|---|---|---|---|
| GEBCO 2014 | Global topography and bathymetry | 30 arc-second | *(Weatherall et al. 2015)* |
| Average soil and sedimentary deposit thickness | A gridded global data set of soil, intact regolith, and sedimentary deposit thicknesses for regional and global land surface modelling, max. estimated depth is 50m | 30 arc-second | (*Pelletier et al.* 2016) |
| PCR-GLOBWB | Thickness of the groundwater layer from the global model (5 arc-minute) | 5 arc-minute | (*de Graaf et al.* 2015) |
| GLIM | Global Lithological Map - Rock types of the Earth surface (16 basic classes), more than 1,200,000 polygons | vector | (*Hartmann & Moosdorf* 2012) |
| Natural Earth coastline | Global coastline | vector | *(Natural Earh, 2017)* |