# Peer review of "Estimating the thickness of unconsolidated coastal aquifers along the global coastline"

_Earth System Science Data, 2017_

## Referee Comment (RC1) · Anonymous Referee #1 · 6 Apr 2018

Review of:
**Estimating the thickness of unconsolidated coastal aquifers along to global coastline**
Zamrsky et al

In this paper a methodology is described to estimate unconsolidated coastal aquifer thickness along the coastal ribbon. The estimation results are validated against borehole data (representing the more local scale) and literature values (representing the regional scale). A numerical experiment, designed to study the changes in salt intrusion under changes in aquifer thickness and vertical structure, was done. The paper concludes that the new method used to estimate the unconsolidated coastal aquifer thickness is suitable at the global-to-regional scale and that geological complexity plays an important role in simulation salinity concentration profiles.

This work is clearly a step in the right direction, and we need to test our ability to simulate these systems accurately. However, currently the manuscript suffers from unintended confusion throughout the whole text; methods, including assumptions made and uncertainties, and results need to be better explained, and I recommend to give the grammar and wordiness an extra check, as well as the clarity of the text. Nevertheless, I think the concerns I have, although it is a long list, can be solved (major revisions).

**Specific concerns:**
1.  My first concern is related to main motivation of this research, which I found not so convincing and is not helping to fully understand to usefulness of the work done. After reading the introduction it is still not so clear why we need a new coastal aquifer dataset and what this new dataset actually contains. For example, in the introduction (p2 L11) "open source global datasets"; can you be more specific here, what kind of datasets? Datasets that hold hydrogeological information? Same sentence "during the estimation process"; estimation of what? Be more specific. Also, previously published datasets are hardly described, which makes it particularly hard to judge the importance of this research. Could you add one or two lines to describe the major lacks of these datasets and why they cannot be used to determine the vulnerability of coastal aquifers to sea-level rise and groundwater pumping, and thus motivate why you estimate coastal aquifer thicknesses and not use e.g. Pelletier's soil data or de Graaf's aquifer thickness estimate.

    Related to the introduction, also in the conclusion part of the paper (last paragraph) I have the feeling the 'take-home message' can be much stronger formulated. What is the big selling point of the data, what is the major improvement compared to previous datasets, how can it be used in global scale estimates of aquifer structure, what did we learn from the numerical experiments and how will this help to improve the current hydrological large-scale models etc.

2.  My second concern is related to the methods, and the many questions I still have after reading these sections. The assumptions and uncertainties of your methodology are not well described nor discussed. For example, it is stated the dataset is limited to unconsolidated sediments only (p2. l11); is this a reasonable assumption, how much of the global coastal ribbon consist of unconsolidated sediments and can you indicate regions where you most likely miss aquifers? And vertically; is this a reasonable assumption considering extensive coastal aquifer systems that may consist of unconsolidated sediments on top of sedimentary rocks (also part of the aquifer system)? Can you say something about the uncertainty in GLiM, as this dataset limits your estimate to the regions classified as unconsolidated sediments in their dataset. How sensitive is you estimate for the placement of the anchor point, and what is the uncertainty of the Pelletier dataset you used to place this anchor point?

    Second, I am a bit confused about p2.L29: "the first study ….estimate the thickness of unconsolidated sediment formations at the global scale"; but I assume unconsolidated sediments of coastal aquifers are meant here. A consistent terminology is not used throughout the text, causing confusion. Besides, I understood the thickness estimation is limited to profiles along the coastal ribbon (like presented in F1d, and with points of F4), but this sentence suggests a spatial distributed estimate of aquifers in general. Can this be clarified?

    Third, I am surprised that you only had 112 borehole descriptions, and I was even more surprised that this does not include any information of the US or Europe; the two continents where normally the most data are available (P4 L16-17). You state the dataset is far from complete, but can you nevertheless explain shortly why US and Europe do not have any borehole data (for example USGS borehole data is freely available as well). Second, how much do you trust your validation if it is only based on so few boreholes.

Additionally, could you provide any insight on taking half of the maximum thickness if no average value was given in your literature review (p4 L26-30)? Does this seem to be a plausible assumption when compared to the reported minimum values you do have in your validation dataset?

Forth, EQ(1) the "relative error percentage" does not exist; what you mean is the percentage relative error (preferable abbreviated as PRE), or percentage error (personally I would stick to the relative error). Also, you need to use brackets for the absolute error here (same in EQ 5), and normally going from a fraction to a percentage is written like this: *relative error X 100%= …%*. Note that the RE of eq. 1 can be negative or positive (please explain the difference) and that in section 3.2 you refer to eq. 1 as calculating the absolute relative error, which is not correct as it is.

Fifth, concerning the "groundwater flow and salt transport modelling"; this section is very short and lacks in motivating the reason why you did this numerical experiment (to study the changes in salt intrusion under changes aquifer thickness and vertical structure in order to make recommendations for future (large-scale) hydrological simulations?). Also, I did not understand what is meant with "different models" p5 l.25; where different models run, or the same model with various parameter settings? I also did not find "the three example cross-sections" L 27.

3. My third concern is related to the discussion of the results. The many errors made in this section make it hard to judge the value of the new findings. For example, P6 L6 "range between 0.1 and more than 5000m": the figure shows a range between 0 and 6500 (map) and 50- More (histogram) (unfortunately a unit is missing in the figure, but I assume meters). And, can you show the distribution of literature and boreholes as well in the histogram?

   Second, throughout this result section and corresponding figures, as well as in section 4: at p2 L24 you say "aquifer thickness estimation method (ATE)". In this section and figures it seems that ATE stands for estimates aquifer thickness; this is confusing.

   Third, P7 L2-7: move this to methodology. Also, an explanation on how you included a more complex geology is needed here. How did you estimate e.g. the location of the aquitard, thickness of the aquitard, and what did you assume for conductivity etc. What are the assumptions and uncertainties in this estimate, and how does this effect the results? Related to the latter, P8 L20- 23: Can you expand on this a bit, maybe reflect on previous studies studying coastal aquifers and salt intrusion that, as far as I know, often simplify the vertical structure of coastal aquifers to a confining layer overlying a confined unconsolidated sediment aquifer.

   Forth, how did you choose your different layering scenarios, and, based on my curiosity, could you say something about the thinnest thickness and the thickest thickness in relation to the previous published thickness estimates. Does the thinnest thickness correspond to what we would get if we used Pelletier's dataset and the thickest if we used De Graaf's dataset for the modelling?

**Minor comments**
- Please check the referencing of the Pelletier dataset:
  p2 L13 the thickness of soil (or regolith) layer;
  p2 L22 the surface sediments
  p2 L27 a thickness of subsurface sediments (regolith)
  P3 L13-14: the soil and sedimentary deposit thickness
  Etc.
- Reconsider the title of section 2.1 e.g. coastal aquifer unconsolidated sediment thickness estimation
- P2 L23: "aquifer thickness estimation from a global hydrological model" : as far as I know a hydrological model does not estimate an aquifer thickness. The estimated aquifer thickness is a parameter in the hydrological model. Just refer to the corresponding study (e.g global scale aquifer thickness estimated by de Graaf et al (2015)), also changes PCR-GLOBWB in the other parts of the text and in Figure 1.
- F1a: If you use references for the Pelletier dataset use references for the others as well (Aquifer thickness, GLiM). Additionally, note that sedimentary thickness is definitely not included in Pelletier. F1b and c: note that the color if the unc.sediments is flipped here.

F1c: it is not clear what the number are in this figure. You mention the second-order estimation line, but which one is this? Overall this figure would benefit from reducing the text per sub-figure (why not reduce the text and move to the figure caption).

- P3, L20-23: are the "four different estimation methods" the ATE method?
- P5 L7 "overall average thickness" is this a global average thickness?
- P6 L7: "A similar result" : A slightly different result
- P6 L8 "literature … 65% … lower than or equal to 300m"
  P6 L19 "literature …. 69% sediments of 300m or lower"
  I would expect the same percentages; can you explain this?
- F5 can the borehole and literature figures somehow be combined, so we can see the differences better? The insets on b and c can be left out.
- P6 L28: There is no figure 3c.
- The Figure 6 that is referred to in section 3.3. should be figure 7, same in section 4. (and the current Figure 6 is not used at all).
- P7 L26: "aquifer thickness (lowest vs highest)" : thinnest vs thickest
- P8 L10: You mention the 200km as a limitation, but at P3 L3-12 you state that this is a correct assumption. In the end not so well chosen? Can you explain this.
- P8 L15 "resolution of the input data" Not exactly clear which input data you mean, also you did not mention the resolution of the input data before.
- P8 L25: you did not mention these studies before.

Examples of unnecessary wordiness from the abstract:
"the thickness of the aquifers" : aquifer thicknesses
"the management and control of groundwater resources" : Groundwater resources management and control
"we must have better information" : we need better information
"the thickness of aquifers" : aquifer thicknesses

---

## Referee Comment (RC2) · Anonymous Referee #2 · 19 May 2018

I think Zamskry et al provide a useful global hydrogeology dataset that if the significant suggestions that reviewers have, could be a worthwhile contribution to ESSD.

I first read the paper to gather my thoughts and then read the comments of reviewer RC1. just to be efficient rather than re-iterating verbosely, I will first say that i largely agree with many of the overall and specific comments of RC1 and hope the authors can and will address of all these comments.

I add a few additional suggestions:

the anchor points are important but hard to know how to interpret - i suggest possibly adding a graph of distance of anchor point to shoreline (histogram or boxplot against lithology might also be interesting) - what is controlling this distance?

[Figure]

similarly, I found the 'four different estimation methods' important but hard to visualise and interpret - could these be shown on a seperate graphic or labeled seperately on Figure 1? Also, these methods are fine mathematically but i was struck by the question: is there not coastal erosion or geomorphology theory/model/observations that would help determine which method is most likely or better. I am thinking of bedrock fluvial enviroments where there is well recognized theory/model/observations that predict river concavity, elevation etc. is there anything similar for coastal erosion?

I also wonder if the authors could analyse and report where the coastal aquifer thickness is zero or effectively zero (<5 m or some other cutoff?). it would be interesting to groundtruth these results against remote sensed information of exposed bedrock if possible.
* * *

---

## Author Comment (AC1) · 15 Jun 2018

Dear reviewers,

We would like to thank you for your valuable input and insightful comments, the vast majority was implemented in the second version of our manuscript. Please find in the supplement the answer to your reviews itself, a clean second version of the manuscript and also the track changes document (compared to the first version) for easier orientation in the changes made to the manuscript.

Once more thank you very much for your time and input.

Daniel Zamrsky, Gu Oude Essink and Marc Bierkens

[Figure]

Please also note the supplement to this comment:
https://www.earth-syst-sci-data-discuss.net/essd-2017-110/essd-2017-110-AC1-
supplement.zip

—————————————————

**[ESSDD](https://www.earth-syst-sci-data-discuss.net)**

Interactive
comment